# The Effect of Ecophysiological Traits on Live Fuel Moisture Content

**Alexandria L. Pivovaroff** [1,*,†] **, Nathan Emery** [2] **, M. Rasoul Sharifi** [3] **, Marti Witter** [4] **, Jon E. Keeley** [5] **and Philip W. Rundel** [3]

[1] La Kretz Center for California Conservation Science, University of California, 619 Charles. E. Young Drive East, Los Angeles, CA 90095, USA

[2] Department of Plant Biology, Michigan State University, 612 Wilson Road, East Lansing, MI 48824, USA; emeryna1@msu.edu

[3] Department of Ecology and Evolutionary Biology, University of California, 612 Charles E. Young Drive South, Los Angeles, CA 90095, USA; sharifi@biology.ucla.edu (M.R.S.); rundel@ucla.edu (P.W.R.)

[4] National Park Service, Santa Monica Mountains National Recreation Area, 401 West Hillcrest Drive, Thousand Oaks, CA 91360, USA; Marti_Witter@nps.gov

[5] United States Geological Survey, Western Ecological Research Center, Sequoia-Kings Canyon Field Station, 47050 Generals Highway, Three Rivers, CA 93271, USA; jon_keeley@usgs.gov

[*] Correspondence: alexandria.pivovaroff@pnnl.gov; Tel.: +1-562-881-4640

[†] Current Address: Atmospheric Sciences and Global Change Division, Pacific Northwest National Lab, 902 Battelle Blvd, Richland, WA 99354, USA.

**Abstract:** Live fuel moisture content (*LFMC*) is an important metric for fire danger ratings. However, there is limited understanding of the physiological control of *LFMC* or how it varies among co-occurring species. This is a problem for biodiverse yet fire-prone regions such as southern California. We monitored *LFMC* and water potential for 11 native woody species, and measured ecophysiological traits related to access to water, plant water status, water use regulation, and drought adaptation to answer: (1) What are the physiological mechanisms associated with changes in *LFMC*? and (2) How do seasonal patterns of *LFMC* differ among a variety of shrub species? We found that *LFMC* varied widely among species during the wet winter months, but converged during the dry summer months. Traits associated with *LFMC* patterns were those related to access to water, such as predawn and minimum seasonal water potentials (Ψ), and water use regulation, such as transpiration. The relationship between *LFMC* and Ψ displayed a distinct inflection point. For most species, this inflection point was also associated with the turgor loss point, an important drought-adaptation trait. Other systems will benefit from studies that incorporate physiological mechanisms into determining critical *LFMC* thresholds to expand the discipline of pyro-ecophysiology.

**Keywords:** live fuel moisture content; water potential; fire; pyro-ecophysiology; chaparral; water relations; functional traits

---

## 1. Introduction

Live fuel moisture content (*LFMC*) is a landscape-level management metric that, along with weather and topography, is incorporated into rate-of-spread models and fire danger ratings [1–4]. *LFMC* is expressed as the ratio of water content in fresh plant tissue to the dry weight and represents the amount of moisture that needs to evaporate from a fuel source before ignition can occur. Greater fuel moisture means reduced flammability and lower likelihood of ignition [5], hence *LFMC* has a major effect on combustion, fire spread, and fire consumption [6–8]. Yet, the relationship between *LFMC* and fire risk is not always linear. Instead, there are thresholds. For example, wildfire risk increases as *LFMC*

decreases until a critical threshold is reached, and then the fire risk is constant and high [9]. However, recent studies suggest that these breakpoints need to be considered with caution as field-based and laboratory-based studies often show differing results related to the magnitude of the effect of *LFMC* on fire rate-of-spread [2,4,10]. Hence, further *LFMC* studies are needed to resolve these issues.

California's Mediterranean-type climate shrublands are especially prone to wildfires. For example, the natural fire regime in southern California historically included infrequent yet large, high-intensity fires [11]. However, modern fire frequencies are higher than normal, which threaten chaparral diversity [12,13]. For chaparral shrubs in southern California, *LFMC* is normally high during the winter and spring and then gradually declines with the onset of the characteristic summer drought (Figure 1) of Mediterranean-type climates (Figure 2a). This leads to a typical fire season about six months long (Figure 1). However, severe or extended drought conditions, which are becoming more common with climate change, lead to fuels drying out sooner, more quickly, to a greater extent, or for longer. In these instances, the predicted fire season starts earlier in the year and/or lasts longer. Extended droughts have been associated with high fire activity or anomalous seasonality. For example, five of nine historic "mega fires" (>50,000 ha) in southern California occurred during a period of anomalous drought [14]. Large, wind driven, spring fires can occur when *LFMC* is at levels that normally occur in the late summer or fall [15].

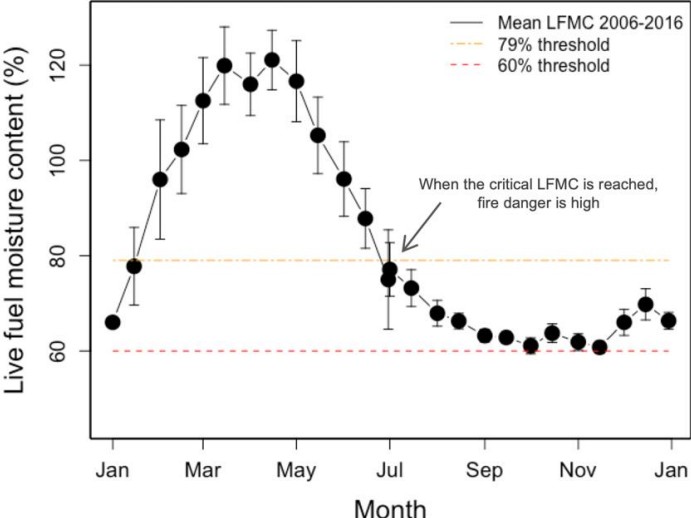

**Figure 1.** Mean live fuel moisture content (*LFMC*) ± standard error, measured for *Adenostoma fasciculatum* at Stunt Ranch Santa Monica Mountains Research Reserve from 2006–2016, represented by the solid black symbols and line. The red dashed line indicates the 60% critical *LFMC* threshold used by local fire departments, while the orange dash-dotted line indicates the 79% threshold from Dennison and Moritz (2009).

In some cases, extreme drought can lead to vegetation mortality [16] and hence increased dead fuel loads, which can also exacerbate fire risk. These drought conditions can lead to increased dead fuels, which are largely controlled by weather conditions and fuel thickness [8]. Alternatively, live fuel moisture covaries with environmental conditions, including temperature, precipitation, and soil moisture, as well as the phenological and physiological characteristics of a plant species [9,17,18]. Despite the widespread use of *LFMC* as a proxy for flammability and fire risk, only recently have quantitative studies emerged of how these data relate to traits of plant water relations components including water potentials, turgor loss point, relative water content, and hydraulic conductivity [19,20]. This represents a significant gap in our understanding of pyro-ecophysiology [21].

Determining the physiological mechanisms that underpin *LFMC*, especially in a fire-prone landscape such as California, has important implications for fire-risk management practices [22]. The objective of this research was to address: (1) What are the physiological mechanisms associated with

changes in *LFMC*? and (2) How do seasonal patterns of *LFMC* differ among a variety of species? To address these questions, we measured *LFMC* and a variety of ecophysiological traits for 11 native woody chaparral and coastal sage scrub species. These traits included water potentials, gas exchange characteristics, and hydraulic traits. We hypothesized that the following traits would control changes in *LFMC*: (1a) traits associated with access to water, such as predawn water potentials and minimum seasonal water potentials, (1b) water use regulation, such as transpiration, and (1c) traits associated with drought adaptation, as determined from pressure-volume curves. We also hypothesized that: (2) these trait differences would manifest as differing seasonal *LFMC* patterns among species.

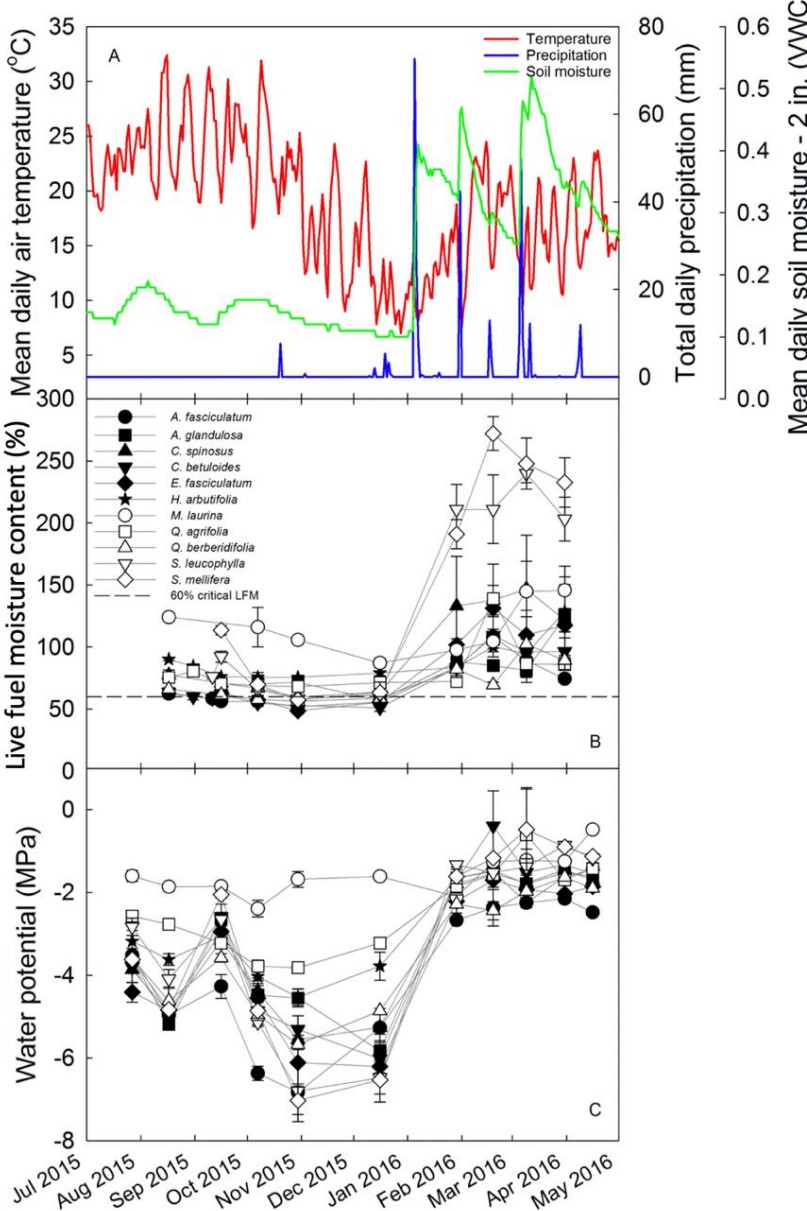

**Figure 2.** (**A**) Weather data for July 2015 to May 2016, including mean daily air temperature (red line), total daily precipitation (blue line), and mean daily soil moisture (green line), measured from the Stunt Ranch Santa Monica Mountains Research Reserve weather station. (**B**) Live fuel moisture content including the 60% critical threshold used by Los Angeles County Fire Department and (**C**) midday water potential measured for 11 species at Stunt Ranch.

## 2. Methods and Materials

### 2.1. Study Site and Species

The study was conducted from July 2015 to March 2016 at the Stunt Ranch Santa Monica Mountains Reserve, located in the Santa Monica Mountains in California (34°5′27″ N, 118°39′27″ W). Stunt Ranch is 125 ha field station dominated by coverage of chaparral, with elevation ranging from 392 to 472 m. With a Mediterranean-type climate, average temperature highs are 32 °C and average lows are 4 °C (Figure 2A). Mean annual precipitation is 610 mm, occurring mostly during the winter months with almost no rainfall during the summer; however, there is very high interannual rainfall variability. Stunt Ranch last burned in 1993, 22–23 years before this study. This study measured 11 common woody species in California shrub land ecosystems (Table 1).

### 2.2. Live Fuel Moisture Content

*LFMC* (%) was measured approximately every three weeks, concurrently with plant water potentials (see below). During sampling, a terminal shoot was collected from each individual, for at least three individuals per species during midday (11:00 to 13:00), approximating the procedures used by the Los Angeles County Fire Department and the Ventura County Fire Department in their monitoring of *LFMC*. Each sample was sealed in a separate plastic bag (Whirlpak, Nasco, and Fort Atkinson, WI, USA) and transported in a cooler to the lab.

*LFMC* is the ratio of water weight to dry weight of living plant tissue [23] and was determined by measuring the sample fresh weight and then placing the sample in a coin envelope and drying in an oven at 70 °C for at least 72 h, after which the dry mass was recorded. *LFMC* (%) was calculated as:

$$LFMC = \frac{Fresh\ weight - Dry\ weight}{Dry\ weight} * 100 \tag{1}$$

Minimum seasonal *LFMC* (*LFMC*$_{MIN}$) was determined from the lowest *LFMC* value a species experienced over the course of a year, and maximum seasonal *LFMC* (*LFMC*$_{MAX}$) was determined from the highest *LFMC* value a species experienced over the course of a year.

### 2.3. Plant Water Potential

At the same time, and from the same branch used for *LFMC* sample collection (see above), an additional terminal shoot was collected for a paired water potential (Ψ; MPa) measurement, for at least three individuals per species during midday (11:00 to 13:00). Each sample was sealed in a separate plastic bag (Whirlpak, Nasco, and Fort Atkinson, WI, USA) and transported in a cooler to the lab. Water potential was determined with a Scholander-type pressure chamber (PMS Instrument Corp., Corvallis, Oregon). In addition to midday water potentials, predawn water potentials (04:00 to 05:00) were sampled during the dry (August 2015; Ψ$_{PD;dry}$) and wet season (March 2016; Ψ$_{PD;wet}$) following the same protocol. Minimum seasonal water potential (Ψ$_{MIN}$) was determined from the most negative water potential a species experienced over the course of a year.

### 2.4. Gas Exchange

Photosynthetic gas exchange was measured using a portable photosynthesis system (LI-6400, LI-COR Biosciences, Lincoln, NE, USA) with a red-blue light source. Measurements were frequently taken from September 2014 to April 2016, during the morning and midday hours (9:00 to 14:00) on the most recently mature leaf. We determined the light-saturated rate of photosynthetic $CO_2$ assimilation per area ($A_{MAX}$; μmol·m$^{-2}$·s$^{-1}$), stomatal conductance to water vapor ($g_S$; mol·m$^{-2}$·s$^{-1}$), transpiration ($E$; mol·m$^{-2}$·s$^{-1}$), and intrinsic water use efficiency ($A/g_S$; μmol·mol$^{-1}$).

## 2.5. Pressure-Volume Curves

Pressure-volume curves were determined on six terminal shoots per species using the bench-dehydration method [24,25] during August and September, which are dry summer months. Samples were collected from the field, sealed in plastic bags, and placed inside a cooler for transport back to the lab. In the lab, shoots were recut under water and placed in beakers with the distal cut stem end in water, and the entire sample covered in plastic to rehydrate overnight. The next day, samples were weighed and measured for water potential using a Scholander-type pressure chamber, and then dried on a lab bench. Weighing, measuring water potential, and drying samples were repeated until achieving a water potential of about −4 to −6 MPa, depending on the species. Subsequently, samples were put in a paper bag and dried in an oven at 70 °C for at least 48 h to determine dry mass. We calculated saturated water content ($SWC$; %), water potential at turgor loss point ($\Psi_{TLP}$; MPa), relative water content at turgor loss point ($RWC_{TLP}$; %), osmotic potential ($\pi_o$; MPa), modulus elasticity ($\varepsilon$; MPa), capacitance at full turgor ($C_{FT}$; MPa$^{-1}$), capacitance at turgor loss point ($C_{TLP}$; MPa$^{-1}$), and leaf dry matter content ($LDMC$; g·g$^{-1}$) using the Excel spreadsheet available on PrometheusWiki [26].

## 2.6. Statistical Analyses

All statistical tests were performed in R Studio (ver 0.99.485) [27]. Piecewise regression using the 'segmented' package was used to find the inflection point, standard error, and 95% confidence intervals for the relationship between water potential and $LFMC$ ($LFMC_{IP}$) for each individual species and for all species combined. We tested for differences in $\Psi_{TLP}$ among species using ANOVA with the 'aov' function from the 'stats' package and a Tukey post-hoc test. Pearson correlation was performed with the 'rcorr' function from the 'Hmisc' package to produce a correlation matrix and evaluate the relationships among all measured traits [28,29]. Before running correlations, traits were tested for normal distribution using the Shapiro-Wilk test of normality [30] and transformed if necessary. Transformed traits included $A_{MAX}$, $g_S$, $E$, $SWC$, and $LFMC_{MAX}$. The correlogram was constructed using the 'corrplot' function.

## 3. Results

$LFMC$ and water potentials ($\Psi$) varied among species and seasons (Figure 2B,C). $LFMC$ was highest and most variable among species during the wet months of January through April, while species tended to converge towards low $LFMC$ during the dry season. $LFMC_{MIN}$ occurred in October and November, while $LFMC_{MAX}$ occurred in February and March. $\Psi$ were lowest during the dry season, particularly in October, November, and December, right before winter rains began. Variability in $\Psi$ among species was greatest during the dry season. When winter rains started in January, species converged towards less negative $\Psi$.

*Salvia mellifera* had the highest $LFMC_{MAX}$, while *Quercus berberidifolia* had the lowest $LFMC_{MAX}$. *Eriogonum fasciculatum* had the lowest $LFMC_{MIN}$, while *Malosma laurina* had the highest $LFMC_{MIN}$. *Salvia mellifera* had the lowest $\Psi_{MIN}$, and *Malosma laurina* had the highest $\Psi_{MIN}$. In fact, deeply rooted *Malosma laurina* did not show a substantial seasonal change in $\Psi$ or $LFMC$.

In comparing $LFMC$ to $\Psi$ (Figure 3, Supplemental Figure S1), $LFMC$ initially decreased rapidly with only a small decline in $\Psi$, before reaching an inflection point ($LFMC_{IP}$) and leveling off. After $LFMC_{IP}$, there were large declines in $\Psi$ with only a small decline in $LFMC$. In addition, the confidence intervals for $LFMC_{IP}$ overlapped with the confidence intervals for $\Psi_{TLP}$ for the compiled data set (Figure 3), which included all study species, as well as for individual species, except for *A. fasciculatum*, *Heteromeles arbutifolia*, *M. laurina*, and *S. leucophylla* (Supplemental Figure S1). For *H. arbutifolia* and *M. laurina*, the relationship between $LFMC$ and $\Psi$ was not tightly constrained, which made it difficult for the model to converge on an inflection point. For *A. fasciculatum*, $\Psi_{TLP}$ was more negative than the inflection point, while for *S. leucophylla*, $\Psi_{TLP}$ was less negative than the inflection point. For the

remaining species, *E. fasciculatum* had the lowest $LFMC_{IP}$ at 67%, while *C. spinosus* had the highest $LFMC_{IP}$ at 121%.

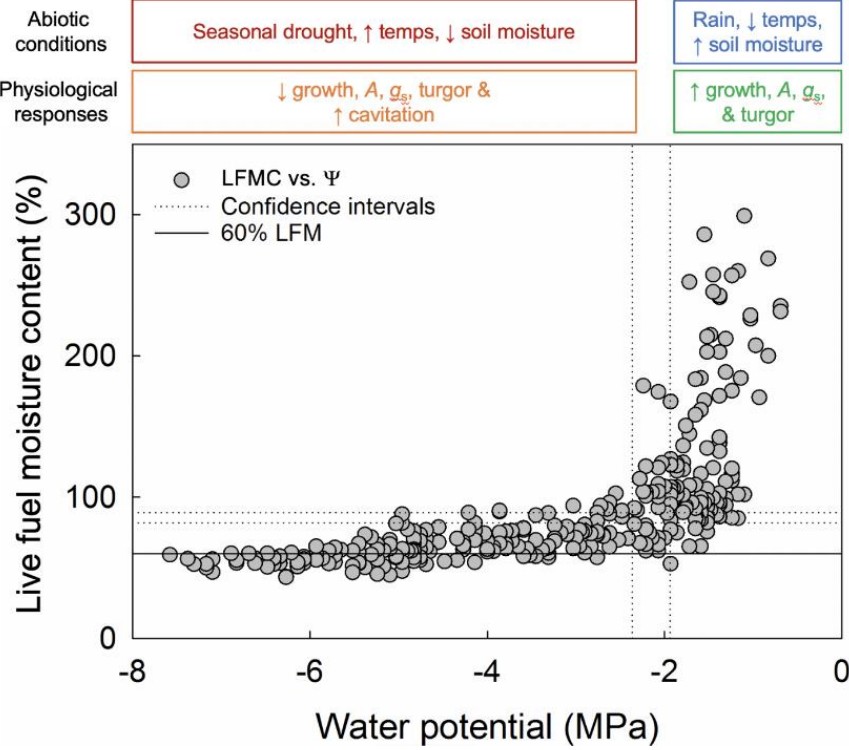

**Figure 3.** Live fuel moisture content (*LMFC*) versus water potential (Ψ) for 11 species measured at the Stunt Ranch Santa Monica Mountains Research Reserve from July 2015–May 2016 (grey symbols). Each grey symbol represents a data point. The solid black line represents the 60% critical *LFMC* threshold used by fire departments in southern California. Dotted lines represent the 95% confidence intervals for Ψ and *LFMC*; the intersection of the confidence intervals represents the *LFMC* versus Ψ inflection point (*LFMC*$_{IP}$). At the top of the figure, there is a conceptual explanation of abiotic conditions and physiological responses associated with *LFMC* and Ψ changes.

Species with lower $LFMC_{MIN}$ also had lower (more negative) $Ψ_{MIN}$, lower $Ψ_{PD;wet}$, and lower $Ψ_{PD;dry}$ (Figure 4). For example, *A. fasciculatum* had the second lowest $Ψ_{MIN}$ and the lowest $Ψ_{PD;wet}$ and $Ψ_{PD;dry}$. $LFMC_{MAX}$ was correlated with $A_{MAX}$, *E*, *SWC*, $C_{FT}$, $C_{TLP}$, and *LDMC*. $LFMC_{IP}$ was correlated with *SWC*.

$Ψ_{TLP}$ among species ranged from −1.47 MPa to −3.40 Mpa (Table 1). *S. leucophylla* had the highest (least negative) $Ψ_{TLP}$ and *A. fasciculatum* had the lowest (most negative) $Ψ_{TLP}$. While *S. leucophylla* had the lowest $Ψ_{TLP}$, it was not significantly different from *Q. berberidifolia*, *C. betuloides*, *M. laurina*, or *S. mellifera*. While *E. fasciculatum* and *A. fasciculatum* had the two lowest (most negative) $Ψ_{TLP}$ and were also significantly different from the five previously mentioned species, they were not significantly different from *A. glauca*, *Q. agrifolia*, *H. arbutifolia*, and *C. spinosus*.

**Table 1.** Study species, including family, leaf phenology, post-fire regeneration type, the live fuel moisture content inflection point (*LFMC*$_{IP}$) ± standard error, and the turgor loss point ($\Psi_{TLP}$; MPa) ± standard error at the Stunt Ranch Santa Monica Mountains Reserve. Sources of post-fire regeneration type: [31][1], [32][2], [33][3]. For $\Psi_{TLP}$, different letter superscripts indicate significant differences (*p* < 0.05).

| Species | Family | Leaf Phenology | Post-Fire Regeneration Type | *LFMC*$_{IP}$ | $\Psi_{TLP}$ |
|---|---|---|---|---|---|
| *Adenostoma fasciculatum* | Rosaceae | Evergreen | Facultative sprouter [2] | 77% ± 5 | −3.40 ± 0.17 [B] |
| *Arctostaphylos glandulosa* | Ericaceae | Evergreen | Facultative sprouter [2] | 86% ± 4 | −2.51 ± 0.27 [B,C] |
| *Ceanothus spinosus* | Rhamnaceae | Evergreen | Facultative sprouter [1] | 121% ± 19 | −2.80 ± 0.29 [B,C] |
| *Cercocarpus betuloides* | Rosaceae | Evergreen | Obligate sprouter [2] | 71% ± 5 | −2.02 ± 0.09 [A,C] |
| *Eriogonum fasciculatum* | Polygonaceae | Semi-deciduous | Reseeder [3] | 67% ± 5 | −3.23 ± 0.20 [B] |
| *Heteromeles arbutifolia* | Rosaceae | Evergreen | Obligate sprouter [2] | 98% ± 14 | −2.70 ± 0.21 [B,C] |
| *Malosma laurina* | Anacardiaceae | Evergreen | Facultative sprouter [2] | 92% ± 10 | −2.07 ± 0.14 [A,C] |
| *Quercus agrifolia* | Fagaceae | Evergreen | Obligate sprouter [2] | 93% ± 12 | −2.61 ± 0.27 [B,C] |
| *Quercus berberidifolia* | Fagaceae | Evergreen | Obligate sprouter [2] | 81% ± 4 | −1.97 ± 0.22 [A,C] |
| *Salvia leucophylla* | Lamiaceae | Drought-deciduous | Reseeder [3] | 85% ± 4 | −1.47 ± 0.11 [A] |
| *Salvia mellifera* | Lamiaceae | Drought-deciduous | Reseeder [3] | 116% ± 8 | −2.08 ± 0.14 [A,C] |

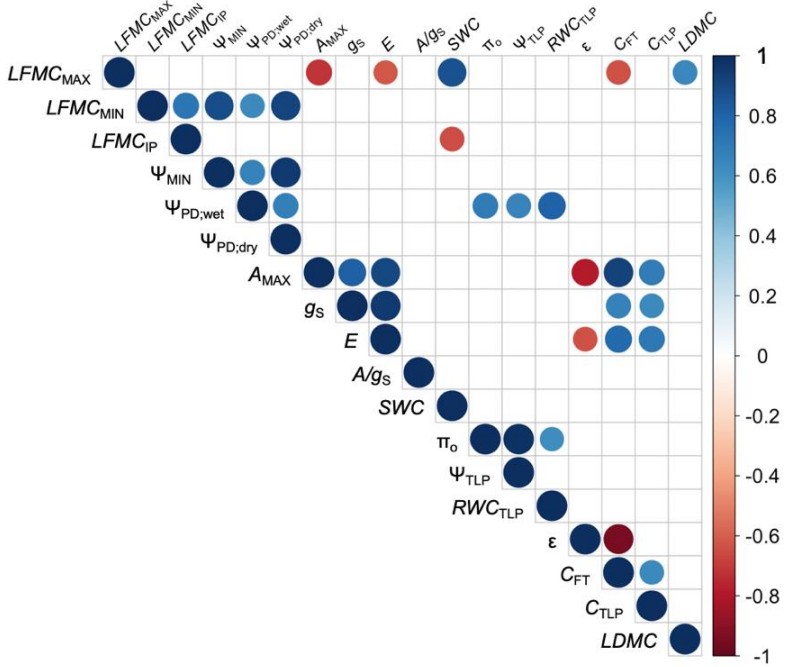

**Figure 4.** Correlogram of all traits measured for 11 species at the Stunt Ranch Santa Monica Mountain Research Reserve from July 2015–May 2016. Blue indicates a positive correlation while red indicates a negative correlation. Blank squares indicate the correlation was not significant at $\alpha = 0.05$. Abbreviations and units are found under the "Abbreviations" table.

## 4. Discussion

### 4.1. Overview

Our study aimed to answer, (1) What are the physiological mechanisms associated with changes in *LFMC*? We found that minimum seasonal live fuel moisture content ($LFMC_{MIN}$) was correlated with water access traits, including wet season predawn water potential ($\Psi_{PD;wet}$), dry season predawn water potential ($\Psi_{PD;dry}$), and minimum seasonal water potential ($\Psi_{MIN}$), confirming hypothesis 1a as well as previous work by [19]. Maximum seasonal live fuel moisture content ($LFMC_{MAX}$) was correlated with photosynthesis ($A_{MAX}$) and transpiration ($E$), confirming hypothesis 1b. $LFMC_{MAX}$ was correlated with drought-adaptation traits, including *SWC*, relative $C_{FT}$, and relative capacitance at zero turgor ($C_{TLP}$), *LDMC*, supporting hypothesis 1c.

We also asked, (2) How do seasonal patterns of *LFMC* differ among a variety of species? We found that species trait differences were associated with differing seasonal *LFMC* patterns among species, confirming hypothesis 2. *LFMC* varied among species to the greatest extent during the wet winter months, but all species converged on low *LFMC* values during the dry season. $LFMC_{MAX}$ occurred during the wet season, along with maximum rate of $A_{MAX}$ and $E$, while $LFMC_{MIN}$ and $\Psi_{MIN}$ occurred during the dry season. The coastal sage scrub species *S. mellifera* and *S. leucophylla*, which are drought-deciduous and shallow-rooted, had much higher $LFMC_{MAX}$ values compared to any other species during the wet season. However, *Salvia* species displayed the lowest $\Psi_{MIN}$. *Malosma laurina*, which is an evergreen chaparral species characterized by deep roots, maintained the highest *LFMC* during the dry season, never dropping below 87%.

### 4.2. Physiological Controls of LFMC

Traits associated with water access and water use regulation were critical determinants of *LFMC* among species. One of the most important traits that controlled *LFMC* in this semi-arid plant community was water potential. Predawn water potentials and minimum seasonal water potentials, which are indicative of species access to water [34] and regulation of water loss when stomata are

closed (i.e., cuticular conductance) [35], were correlated with $LFMC_{MIN}$ (Figure 4). Deeply rooted plants have greater access to soil moisture for longer into the dry season than shallow-rooted plants, supporting either the delay of *LFMC* decline or keeping minimum *LFMC* high [36]. In addition to access to water, traits associated with water use regulation played a significant role in controlling *LFMC*. For example, species with the highest $LFMC_{MAX}$ also had high $A_{MAX}$ and high *E*, providing evidence that species with high gas exchange rate rates also have high moisture content in their tissue. Overall, this points to a suite of traits that coordinate water access, water use regulation, and plant water status [37].

By plotting concurrently measured *LFMC* and Ψ together on the same plot, we examined the relationship between a landscape-level management metric used for assaying fire danger (*LFMC*) and a well-established physiological measurement of plant water status (Ψ). As plants started to dry out, *LFMC* rapidly declined with only a small change in Ψ, before reaching an inflection point ($LFMC_{IP}$), after which there were large declines in Ψ with only a small change in *LFMC*. The confidence intervals for $LFMC_{IP}$ overlapped with turgor loss point ($Ψ_{TLP}$) confidence intervals for the compiled data set. $Ψ_{TLP}$ is an important drought-adaptation trait; species with a lower (more negative) $Ψ_{TLP}$ can withstand more negative Ψ before losing leaf turgor, or before wilting. At the point that leaves have lost so much water that they lose turgor, they would also be highly flammable as the amount of moisture in the leaves would be low, hence ignition could easily occur. However, this would correspond to a higher critical *LFMC* threshold than the current 60% used by fire departments in southern California. This finding confirms work by Dennison & Moritz [38], which used a different approach to determine a critical *LFMC* threshold involving precipitation records, and builds on the recent *LFMC* versus fire behavior breakpoint work of Pimont [10,39], with our physiological data.

While the $LFMC_{IP}$ corresponded to the $Ψ_{TLP}$ for the compiled data set, this was not the case for each individual species (Supplemental Figure S1). There are a few possibilities for this result. One cause could be phenology [40] and, more specifically, growth. Dry matter "accumulates" when plants grow, which would exert a strong seasonal effect on the relationship between $LFMC_{IP}$ and $Ψ_{TLP}$ for species with differing phenological patterns [41,42]. Another potential factor is related to water storage and water release dynamics of different plant species, including the release of capillary water versus water released from cavitation events [43,44]. Using a physiologically meaningful *LFMC* threshold, as supported by flammability and precipitation data, along with an appropriate indicator species is important when monitoring fire-risk conditions.

### 4.3. Indicator Species Choice Can Impact Fire Danger Rating

As we have shown, *LFMC* and therefore fire-risk monitoring will vary depending on the indicator species. By comparing 11 dominant species in southern California, a region with high plant biodiversity, we can quantitatively compare *LFMC* of *Adenostoma fasciculatum*, the current indicator species, against other co-occurring species. *Adenostoma fasciculatum* is used as an *LFMC* indicator species in southern California because of its widespread distribution and abundance. In fact, it is present in 70% of California chaparral communities [45]. In addition, it is viewed as being highly flammable due to its oily leaves (giving rise to its other common name "greasewood") and multi-stemmed canopy structure that consists of dense clusters of small stems, branches, and flower stalks [46]; about 60% of stems are <1.27 cm in diameter [47]. This makes it a fine fuel with small stems and leaves, especially in comparison to co-occurring large branching or broad-leafed species. Our data show that of the 11 study species examined, *A. fasciculatum* was one of the first species to reach the 79% critical *LFMC* threshold proposed by [38] and it stayed below that threshold during the seasonal dry period. In fact, it often also fell below the 60% *LFMC* threshold used by LACFD when many other species did not. The only other species to consistently do so was *C. betuloides*. Hence, *A. fasciculatum* is a good indicator species for measuring *LFMC* in southern California chaparral mountains. The high seasonal *LFMC* values of *M. laurina*, *H. arbutifolia*, and *Q. agrifolia* make them less optimal as *LFMC* indicator species.

## 5. Conclusions

In their 2018 Perspective paper, Jolly and Johnson [21] argue,

*The discipline of ecophysiology is rich and mostly unleveraged in live fuel research, yet it has the potential to link plant flammability traits at both the leaf and plant level to fundamental laws that govern how plants functions.*

Our study takes the initial steps to address this and shows that traits associated with water access and water use regulation were critical determinants of seasonal *LFMC* patterns among species. Other systems would benefit from physiological studies of their chosen indicator species and critical *LFMC* thresholds. This understanding is crucial as the number and size of large fires in the western United States have increased, especially when coincident with drought [48]. Increasing temperatures and longer, more intense, and/or more frequent droughts will become common with climate change, so accurately forecasting wildfire risk is an immediate issue.

**Supplementary Materials:** The following are available online at http://www.mdpi.com/2571-6255/2/2/28/s1: Figure S1: Water potential versus live fuel moisture content for 11 species, measured at the Stunt Ranch Santa Monica Mountains Reserve, with the piecewise linear regression used to determine the inflection point.

**Author Contributions:** Conceptualization, A.L.P. with input from M.R.S., M.W., J.E.K., and P.W.R. Methodology, A.L.P. Formal analysis, A.L.P. with input from N.E., M.R.S., M.W., J.E.K., and P.W.R. Writing–Original draft preparation, A.L.P. Writing–Reviewing and editing, A.L.P., N.E., M.R.S., M.W., J.E.K., and P.W.R.

**Funding:** This work was supported by funding provided by a La Kretz Center for California Conservation Science Postdoctoral Fellowship to A.L.P. and the Bloom-Hays Ecological Research Grant from Sea and Sage Audubon to A.L.P.

**Acknowledgments:** We thank Kim Riley, Linda Moua, Mindy Phan, and Tai Michaels for help in the field and lab; Aaron Ramirez, the Sork lab, and the Sack lab for useful feedback on the data; Mario Colon and Gary Bucciarelli for logistical support at the Stunt Ranch Santa Monica Mountains Reserve. We acknowledge that this study was conducted on the traditional territory of the Chumash and Tongva people.

**Conflicts of Interest:** The authors declare no conflict of interest.

## Abbreviations

| Trait | Abbreviation | Units |
|---|---|---|
| Live fuel moisture content | *LFMC* | % |
| Minimum seasonal live fuel moisture content | $LFMC_{MIN}$ | % |
| Maximum seasonal live fuel moisture content | $LFMC_{MAX}$ | % |
| Live fuel moisture inflection point | $LFMC_{IP}$ | % |
| Water potential | $\Psi$ | MPa |
| Wet season predawn water potential | $\Psi_{PD;wet}$ | MPa |
| Dry season predawn water potential | $\Psi_{PD;dry}$ | MPa |
| Minimum seasonal water potential | $\Psi_{MIN}$ | MPa |
| Maximum seasonal water potential | $\Psi_{MAX}$ | MPa |
| Maximum photosynthetic carbon gain | $A_{MAX}$ | $\mu mol \cdot m^{-2} \cdot s^{-1}$ |
| Stomatal conductance | $g_S$ | $mol \cdot m^{-2} \cdot s^{-2}$ |
| Transpiration | *E* | $mol \cdot m^{-2} \cdot s^{-2}$ |
| Water use efficiency | $A/g_S$ | $\mu mol \cdot mol^{-1}$ |
| Saturated water content | *SWC* | % |
| Water potential at turgor loss point | $\Psi_{TLP}$ | MPa |
| Relative water content at turgor loss point | $RWC_{TLP}$ | % |
| Osmotic potential | $\pi_o$ | MPa |
| Modulus of elasticity | $\varepsilon$ | MPa |
| Capacitance at full turgor | $C_{FT}$ | $MPa^{-1}$ |
| Capacitance at turgor loss point | $C_{TLP}$ | $MPa^{-1}$ |
| Leaf dry matter content | *LDMC* | $g \cdot g^{-1}$ |

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
