# Peer review of "The Effect of Ecophysiological Traits on Live Fuel Moisture Content"

_fire, doi:10.3390/fire2020028_

Round 1

Reviewer 1 Report

This study examines the relationship between LFMC dynamics and physiological traits (gas exchange, drought related traits such as water potential and pressure volume curves parameters) for different shrub species. The authors found a non-linear relationship between Psi and LFMC for most species. Also they report interesting correlations at the interspecific levels:

1-    between gas exchanges traits (max gs, max A, WUE) and LFMCmax supporting the idea that species that favour high gas exchanges (growth) also a have high moisture content of their living tissue.

2-    between min predawn water potential and LFMCmin. Indicating that species with lower rooting depth (or species less constrain water loss: with higher gmin for instance, see Martin-StPaul et al 2017) tend to experience lower LFMCmin

3-    between some PV curve parameters (LDMC, SWC, Capacitance) and LFMCmax, which indicate that laboratory traits are consistent with field measurements.

However no relationship is found/shown between LFMC at inflection point and Ytlp (which is expected: Tyree & Yang 1990 Planta) which should be discuss. I recall that two processes influence LFMC dynamics (Jolly et al 2014):

Ø  dry matter accumulation related to growth (particularly important at young stages of shoot development)

Ø  water relation related to tissue dehydration/desiccation dictated by the pressure volume curves and the cavitation resistance.

Can it be that the very high value of LFMC obtained were related to the production of new shoots? And thus that the decrease in LFMC at high water potential (the non-linear phase) are related to dry matter accumulation rather than tissue dehydration? In this case, it is normal that PV curves traits can explain much of the rate of change of LFMC with psi. By contrast, changes in phenological stages that could help understand high LFMC values? Did you estimate this? I think that in your case it is important to discuss this patterns or present additional data.

Overall this a nice study and an impressive set of data. However, before the papers being published, I suggest the authors discuss the previous point. I also suggest the authors refer to some of the very recent literature on the topic and account for the following comments:

1-    L46-50. Please have a look and cite the recent the critics brought by Pimont et al 2018a (IJWF) on the use of classical methods to detect break points in LFMC vs fire behaviour. Regarding the role of LFMC on fire behaviour, you could also be interested by Pimont et al 2018b (IJWF) who re-analysed experimental fire data to strengthen the nonlinear effect of LFMC on fire rate of spread.

2-    L80: here you could have look to and cite Ruffault et al 2018 AFM along with Nolan paper. Ruffault et al 2018 suggest that traits (including pressure volume curves and rooting depth) define interspecific variations in drought vs LFMC relationship and help to understand why some species react differently from each other.

3-    L89-91. Actually, minimum water potential (or minimum values of predawn water potential) not only depends on rooting depth but also (and more likely) on minimum conductance when stomata are closed (Martin-StPaul et al 2017 Ecology Letters). You can acknowledge for this somewhere.

4-    L291 regarding the relationship between LFMC and remote-sensing index: You could cite Fan et al 2017 RSE who used remote sensing to predict shoot level LFMC. And Momem et al 2017 JGR who showed correlation between water potential and remote sensing VOD.

5-    L300-301 again, you can account for the critics brought by Pimont et al 2018a (IJWF) on the use of classical methods to detect break points in LFMC vs fire behaviour.

6-    Did you try test the relationship between LFMCtlp and RWCtlp. Epsilon plays also a critical role in RWCtlp.

Figure 4. Could you please indicate the units for the different variable tested, on the plots axis or in the legend.

Best regards,

Nicolas Martin-StPaul

References:

Momen M, Wood JD, Novick KA, Pangle R, Pockman WT, McDowell NG, Konings AG (2017) Interacting Effects of Leaf Water Potential and Biomass on Vegetation Optical Depth. Journal of Geophysical Research: Biogeosciences 122, 3031–3046. doi:10.1002/2017JG004145.

Jolly WM, Hadlow AM, Huguet K (2014) De-coupling seasonal changes in water content and dry matter to predict live conifer foliar moisture content. International Journal of Wildland Fire 23, 480–489. doi:10.1071/WF13127.

Pimont F, Ruffault J, Martin-Stpaul NK, Dupuy JL (2019) Why is the effect of live fuel moisture content on fire rate of spread underestimated in field experiments in shrublands? International Journal of Wildland Fire. doi:10.1071/WF18091.

Ruffault J, Martin-StPaul N, Pimont F, Dupuy JL (2018) How well do meteorological drought indices predict live fuel moisture content (LFMC)? An assessment for wildfire research and operations in Mediterranean ecosystems. Agricultural and Forest Meteorology 262, 391–401. doi:10.1016/j.agrformet.2018.07.031.

Fan L, Wigneron JP, Xiao Q, Al-Yaari A, Wen J, Martin-StPaul N, Dupuy JL, Pimont F, Al Bitar A, Fernandez-Moran R, Kerr YH (2018) Evaluation of microwave remote sensing for monitoring live fuel moisture content in the Mediterranean region. Remote Sensing of Environment 205, 210–223. doi:10.1016/j.rse.2017.11.020.

Pimont F, Ruffault J, Martin-Stpaul NK, Dupuy JL (2019) A Cautionary Note Regarding the Use of Cumulative Burnt Areas for the Determination of Fire Danger Index Breakpoints. International Journal of Wildland Fire 254–258. doi:10.1071/WF18056.

Martin-StPaul N, Delzon S, Cochard H (2017) Plant resistance to drought depends on timely stomatal closure (H Maherali, Ed.). Ecology Letters 1–23. doi:10.1111/ele.12851.

Tyree MT, Yang S (1990) Water-storage capacity of Thuja, Tsuga and Acer stems measured by dehydration isotherms - The contribution of capillary water and cavitation. Planta 182, 420–426. doi:10.1007/BF02411394.

Author Response

REVIEW 1

Comments and Suggestions for Authors

This study examines the relationship between LFMC dynamics and physiological traits (gas exchange, drought related traits such as water potential and pressure volume curves parameters) for different shrub species. The authors found a non-linear relationship between Psi and LFMC for most species. Also they report interesting correlations at the interspecific levels:

1-    between gas exchanges traits (max gs, max A, WUE) and LFMCmax supporting the idea that species that favour high gas exchanges (growth) also a have high moisture content of their living tissue.

2-    between min predawn water potential and LFMCmin. Indicating that species with lower rooting depth (or species less constrain water loss: with higher gmin for instance, see Martin-StPaul et al 2017) tend to experience lower LFMCmin

3-    between some PV curve parameters (LDMC, SWC, Capacitance) and LFMCmax, which indicate that laboratory traits are consistent with field measurements.

However no relationship is found/shown between LFMC at inflection point and Ytlp (which is expected: Tyree & Yang 1990 Planta) which should be discuss. I recall that two processes influence LFMC dynamics (Jolly et al 2014):

Ø  dry matter accumulation related to growth (particularly important at young stages of shoot development)

Ø  water relation related to tissue dehydration/desiccation dictated by the pressure volume curves and the cavitation resistance.

Can it be that the very high value of LFMC obtained were related to the production of new shoots? And thus that the decrease in LFMC at high water potential (the non-linear phase) are related to dry matter accumulation rather than tissue dehydration? In this case, it is normal that PV curves traits can explain much of the rate of change of LFMC with psi. By contrast, changes in phenological stages that could help understand high LFMC values? Did you estimate this? I think that in your case it is important to discuss this patterns or present additional data.

While minimum and maximum seasonal LFMC were not correlated with turgor loss point, the confidence intervals for the inflection point between LFMC and water potential and the confidence intervals for turgor loss point overlapped.  The relationship between LFMC vs. water potential serves as a sort of in situ pressure volume curve.  However, I agree that the LFMC inflection point did not correspond to turgor loss point for each individual species (Supplemental Figure 1).  To address this, we’ve added another paragraph to the Discussion (lines 298-306) that cover the points raised by the reviewer, and reference the indicated studies.  In addition, we have a manuscript under review for a larger study that that spans sites across California that specifically examines the relationship between LFMC and phenology that we did not previously reference:

Emery, N, Roth K, Pivovaroff AL. Flowering phenology is associated with plant flammability. Under review at Ecological Indicators since 25-March-2019.

Overall this a nice study and an impressive set of data. However, before the papers being published, I suggest the authors discuss the previous point. I also suggest the authors refer to some of the very recent literature on the topic and account for the following comments:

1-    L46-50. Please have a look and cite the recent the critics brought by Pimont et al 2018a (IJWF) on the use of classical methods to detect break points in LFMC vs fire behaviour. Regarding the role of LFMC on fire behaviour, you could also be interested by Pimont et al 2018b (IJWF) who re-analysed experimental fire data to strengthen the nonlinear effect of LFMC on fire rate of spread.

Thank you for the references.  They were extremely helpful in rounding out our paper, and we added these studies and corresponding test related to the studies and our current study throughout the Introduction and Discussion.

2-    L80: here you could have look to and cite Ruffault et al 2018 AFM along with Nolan paper. Ruffault et al 2018 suggest that traits (including pressure volume curves and rooting depth) define interspecific variations in drought vs LFMC relationship and help to understand why some species react differently from each other.

We added this reference.

3-    L89-91. Actually, minimum water potential (or minimum values of predawn water potential) not only depends on rooting depth but also (and more likely) on minimum conductance when stomata are closed (Martin-StPaul et al 2017 Ecology Letters). You can acknowledge for this somewhere.

We added the Martin-StPaul et al 2017 Ecology Letters references and edited the associated text here and throughout by removing reference to minimum water potential serving as “a proxy for rooting depth” to reflect that minimum water potentials can also be associated with cuticular conductance. See also lines 277-278 in the Discussion.

4-    L291 regarding the relationship between LFMC and remote-sensing index: You could cite Fan et al 2017 RSE who used remote sensing to predict shoot level LFMC. And Momem et al 2017 JGR who showed correlation between water potential and remote sensing VOD.

With the addition of another paragraph to the Discussion and due to a suggestion from reviewer 3 that the Discussion should actually be shortened, we removed the section of Discussion that previously mentioned remote-sensing of LFMC.

5-    L300-301 again, you can account for the critics brought by Pimont et al 2018a (IJWF) on the use of classical methods to detect break points in LFMC vs fire behaviour.

We again added this reference here and to other spots in the manuscript.

6-    Did you try test the relationship between LFMCtlp and RWCtlp. Epsilon plays also a critical role in RWCtlp.

Based on this comment and another from reviewer 3, we changed Figure 4 to reflect the entire correlation matrix, rather than just the significant relationship which now shows the not significant correlation between LFMCtlp and RWCtlp.

Figure 4. Could you please indicate the units for the different variable tested, on the plots axis or in the legend.

We added the units for the variables tested in both the methods section and in a table at the end of the paper, as also suggested by another reviewer.

Best regards,

Nicolas Martin-StPaul

References:

Momen M, Wood JD, Novick KA, Pangle R, Pockman WT, McDowell NG, Konings AG (2017) Interacting Effects of Leaf Water Potential and Biomass on Vegetation Optical Depth. Journal of Geophysical Research: Biogeosciences 122, 3031–3046. doi:10.1002/2017JG004145.

We did not add this reference because, as mentioned above, we removed mention of remoting sensing LFMC and water potentials.

Jolly WM, Hadlow AM, Huguet K (2014) De-coupling seasonal changes in water content and dry matter to predict live conifer foliar moisture content. International Journal of Wildland Fire23, 480–489. doi:10.1071/WF13127.

Added; see reference #38

Pimont F, Ruffault J, Martin-Stpaul NK, Dupuy JL (2019) Why is the effect of live fuel moisture content on fire rate of spread underestimated in field experiments in shrublands? International Journal of Wildland Fire. doi:10.1071/WF18091.

Added; see reference #10

Ruffault J, Martin-StPaul N, Pimont F, Dupuy JL (2018) How well do meteorological drought indices predict live fuel moisture content (LFMC)? An assessment for wildfire research and operations in Mediterranean ecosystems. Agricultural and Forest Meteorology 262, 391–401. doi:10.1016/j.agrformet.2018.07.031.

Added; see reference #18

Fan L, Wigneron JP, Xiao Q, Al-Yaari A, Wen J, Martin-StPaul N, Dupuy JL, Pimont F, Al Bitar A, Fernandez-Moran R, Kerr YH (2018) Evaluation of microwave remote sensing for monitoring live fuel moisture content in the Mediterranean region. Remote Sensing of Environment 205, 210–223. doi:10.1016/j.rse.2017.11.020.

We did not add this reference because, as mentioned above, we removed mention of remoting sensing LFMC and water potentials.

Pimont F, Ruffault J, Martin-Stpaul NK, Dupuy JL (2019) A Cautionary Note Regarding the Use of Cumulative Burnt Areas for the Determination of Fire Danger Index Breakpoints. International Journal of Wildland Fire 254–258. doi:10.1071/WF18056.

Added; see reference #37

Martin-StPaul N, Delzon S, Cochard H (2017) Plant resistance to drought depends on timely stomatal closure (H Maherali, Ed.). Ecology Letters 1–23. doi:10.1111/ele.12851.

Added; see reference #33

Tyree MT, Yang S (1990) Water-storage capacity of Thuja, Tsuga and Acer stems measured by dehydration isotherms - The contribution of capillary water and cavitation. Planta 182, 420–426. doi:10.1007/BF02411394.

Added; see reference #40

Reviewer 2 Report

General comment

This manuscript (MS) discusses the effect of plant ecophysiological traits on live fuel moisture content (LFMC). I find the work pertinent, given the implications of LFMC on fire behaviour; in fact, the influence of LFMC on fire behaviour has been the subject of my research work. I am not an expert on plant traits, thus my review will focus on fire behaviour and on the overall aspects of the MS. I trust that the Editorial team has invited other reviewers with specific expertise on plant traits.

My opinion is that the present MS merits publication. Overall, I find it concise as it delivers the message in a short amount of pages, it is well written and reads well. I have only a few specific comments that I hope can help in improving the MS. If you decide to respond to this review, please reply to each comment individually and refer to the lines in the revised manuscript where changes were made, so they can be tracked easily.

Specific comments

Abstract:

- The abstract is rather long: Fire formatting guidelines refer to a maximum of 200 words. Try to focus on the main message you want to deliver and remove less relevant information.

Introduction:

- Ln 43: Here you cite three publications. However, one of them seems to be a conference paper that I could not access, and the other two give little or no support to your statement. There are many publications more appropriate for this case; for example, there are two recent papers published in Fire directly related to this subject:

Rossa CG, Fernandes PM (2018) Live fuel moisture content: The ‘pea under the mattress’ of fire spread rate modeling? Fire 1(3), 43. doi:10.3390/fire1030043

Rossa CG, Fernandes PM (2018) An empirical model for the effect of wind on fire spread rate. Fire 1(2), 31. doi:10.3390/fire1020031

- Lns 64-70: Is Figure 1 data correct? It seems very awkward that historical average LFMC throughout Apr-Dec does not differ significantly from exceptional drought. Also, exceptional drought LFMC only goes below the 60% threshold used by the local fire department in Jan. 

Monthly average LFMC for Mediterranean shrubs in October is 85%; in the first days of Oct-2017 measured LFMC was 50%, a drop of 35%!

Methods and Materials:

- In this section you use a lot of symbols, which make the remainder of the MS difficult to interpret. Please add a ‘List of symbols’ at the end of the MS before the References. It will make it so much easier for the reader…

- Lns 108-109: You mention that at the time of your study, California was experiencing intense drought conditions. What are the implications of this? Does this mean that your results are only applicable to drought years?

Discussion:

- You end the paper with the ‘Discussion’, which is a format sometimes used in papers. But I encourage you to add a ‘Conclusion’, if necessary replacing some of the current contents of the ‘Discussion’. Many readers do not have the time to go through the papers in detail; by adding a section with the main conclusions of our work the papers become more attractive to be cited.

Author Response

REVIEW 2

General comment

This manuscript (MS) discusses the effect of plant ecophysiological traits on live fuel moisture content (LFMC). I find the work pertinent, given the implications of LFMC on fire behaviour; in fact, the influence of LFMC on fire behaviour has been the subject of my research work. I am not an expert on plant traits, thus my review will focus on fire behaviour and on the overall aspects of the MS. I trust that the Editorial team has invited other reviewers with specific expertise on plant traits.

My opinion is that the present MS merits publication. Overall, I find it concise as it delivers the message in a short amount of pages, it is well written and reads well. I have only a few specific comments that I hope can help in improving the MS. If you decide to respond to this review, please reply to each comment individually and refer to the lines in the revised manuscript where changes were made, so they can be tracked easily.

Specific comments

Abstract:

- The abstract is rather long: Fire formatting guidelines refer to a maximum of 200 words. Try to focus on the main message you want to deliver and remove less relevant information.

The abstract has been cut down to 199 words.

Introduction:

- Ln 43: Here you cite three publications. However, one of them seems to be a conference paper that I could not access, and the other two give little or no support to your statement. There are many publications more appropriate for this case; for example, there are two recent papers published in Fire directly related to this subject:

Rossa CG, Fernandes PM (2018) Live fuel moisture content: The ‘pea under the mattress’ of fire spread rate modeling? Fire 1(3), 43. doi:10.3390/fire1030043

Rossa CG, Fernandes PM (2018) An empirical model for the effect of wind on fire spread rate. Fire 1(2), 31. doi:10.3390/fire1020031

The first paragraph of the Introduction has been reorganized and bit and we have edited the references to include those listed above (lines 37-38).  See reference #’s 3 and 4.

- Lns 64-70: Is Figure 1 data correct? It seems very awkward that historical average LFMC throughout Apr-Dec does not differ significantly from exceptional drought. Also, exceptional drought LFMC only goes below the 60% threshold used by the local fire department in Jan. 

We agree this was confusing, as the “historical average” also included the exceptional drought data. To make our point more straightforward, we edited Figure 1.

Monthly average LFMC for Mediterranean shrubs in October is 85%; in the first days of Oct-2017 measured LFMC was 50%, a drop of 35%!

Methods and Materials:

- In this section you use a lot of symbols, which make the remainder of the MS difficult to interpret. Please add a ‘List of symbols’ at the end of the MS before the References. It will make it so much easier for the reader…

We added a table with all traits, symbols, and units to the end of the manuscript before the References.

- Lns 108-109: You mention that at the time of your study, California was experiencing intense drought conditions. What are the implications of this? Does this mean that your results are only applicable to drought years?

We removed this reference related to the drought conditions in California as the time of this study because these results are applicable to all years in California, both wet and dry.

Discussion:

- You end the paper with the ‘Discussion’, which is a format sometimes used in papers. But I encourage you to add a ‘Conclusion’, if necessary replacing some of the current contents of the ‘Discussion’. Many readers do not have the time to go through the papers in detail; by adding a section with the main conclusions of our work the papers become more attractive to be cited.

We made significant edits to the Discussion, including adding sub-titles, and also took some of the information originally in the Discussion and moved it to the new Conclusion section.

Reviewer 3 Report

In general this experiment design was sound to support the conclusion. A few specific comments are as follows:

line 71-81. The studies in plant traits is lacking to understand LFMC variation. Besides water content traits, there are other correlated LFMC shown in other papers. Consider to cite the papers like: Qi, Y., P. E. Dennison, W. M. Jolly, R. C. Kropp, and S. C. Brewer. 2014. Spectroscopic analysis of seasonal changes in live fuel moisture content and leaf dry mass. Remote Sensing of Environment 150:198-206.

Figure 4: The plot does not look like a symmetric correlation matrix.

Line 241-254. No need to repeat the questions and hypotheses. Consider to reorganize the texts by elaborate the evidences to answers those questions. 

line 289-292: Elaborate why remote sensing of water potential is needed, given remote sensing of LFMC is potential but challenging (shown in the cited paper). Any challenge to remotely sense water potential? 

line 313: Font changed. 

Line 322-328. Elaborate why water traits is needed to forecast wildfire risk? What can water traits tell, but LFMC can not? 

Author Response

REVIEW 3

Comments and Suggestions for Authors

In general this experiment design was sound to support the conclusion. A few specific comments are as follows:

line 71-81. The studies in plant traits is lacking to understand LFMC variation. Besides water content traits, there are other correlated LFMC shown in other papers. Consider to cite the papers like: Qi, Y., P. E. Dennison, W. M. Jolly, R. C. Kropp, and S. C. Brewer. 2014. Spectroscopic analysis of seasonal changes in live fuel moisture content and leaf dry mass. Remote Sensing of Environment 150:198-206.

We added this refences on lines 304.  See reference #39.

Figure 4: The plot does not look like a symmetric correlation matrix.

Because of this comment and another brought up by Reviewer 1, we edited Figure 4 to show all results from the correlation matrix, not just the significant results.

Line 241-254. No need to repeat the questions and hypotheses. Consider to reorganize the texts by elaborate the evidences to answers those questions. 

We removed the repetition of the hypotheses, added sub-headings, and re-organized the Discussion a bit to address multiple reviewer comments.

line 289-292: Elaborate why remote sensing of water potential is needed, given remote sensing of LFMC is potential but challenging (shown in the cited paper). Any challenge to remotely sense water potential?

During editing of the Discussion, we removed reference to remote sensing of LFMC and water potentials.

line 313: Font changed. 

We changed the font to match the rest of the manuscript.

Line 322-328. Elaborate why water traits is needed to forecast wildfire risk? What can water traits tell, but LFMC can not? 

We elaborate on why water traits are needed to forecast wildfire risk in our revised “Conclusion” (lines 351-362).